# A true face of Indian married couples: Effect of age and education on control over own sexuality and sexual violence

Gyan Chandra Kashyap[1], Bal Govind[2], Shobhit Srivastava[3]*, Veena R.[4], Madhumita Bango[5], Subhojit Shaw[3]

1 Institute of Health Management Research, Electronic City (Phase-I), Bengaluru, Karnataka, India, 2 Gokhale Institute of Politics and Economics, Deccan Gymkhana, Pune, Maharashtra, India, 3 International Institute for Population Sciences, Deonar, Mumbai, Maharashtra, India, 4 Healthcare Programs, International School of Business and Research, Infosys Drive, Bengaluru, Karnataka, India, 5 School of Health Systems Studies, Tata Institute of Social Sciences, Deonar, Mumbai, Maharashtra, India

* Shobhitsrivastava889@gmail.com

**Data Availability Statement:** The data used in this study are third party data from DHS (https://dhsprogram.com/data/dataset/India_Standard-DHS_2015.cfm?flag=0) and can be accessed

## Abstract

### Introduction

Though there are several interventions evaluated over the past 25 years, significant knowledge gaps continue to exist regarding the effective prevention of sexual violence. This study explored the socio-economic and context-specific distinctive characteristics of husbands and wives on sexual autonomy and unwanted sexual experiences of currently married women in India.

### Methodology

We have utilized the recent round of National Family Health Survey (NFHS-4, 2015–16) data for this exploration. The NFHS-4 survey had adopted a stratified two-stage sample design to reach out to the survey households. A total of 63,696 couples are included in the analysis comprising of women of 15–49 years age and men of 15–54 years age. Multivariate techniques have been applied to understand the adjusted effects of socio-economic and demographic variables on control over their sexuality and sexual violence.

### Results

Uneducated women married to uneducated men experienced more sexual violence and had less control over their sexuality than the other categories. The adjusted multivariate logistic model shows that educated husbands were significantly more likely to exercise control over their educated wives' sexuality (AOR = 0.88; CI:0.78–0.99). Women having older husbands were significantly less likely to be having no-control over own sexuality (AOR = 0.89; CI:0.83–0.95) and experienced sexual violence (AOR = 0.81; CI:0.70–0.95). Women having comparatively more-educated husbands were significantly less likely to experience sexual violence (AOR = 0.62; CI:0.47–0.81). Muslim women were significantly more likely to have no control overown sexuality. SC/ST women were significantly more likely to experience sexual violence (28%).

following the protocol outlined in the Methods section.

**Funding:** The author(s) received no specific funding for this work.

**Competing interests:** The authors have declared that no competing interests exist.

## Conclusions

This study highlights the factors associated with control over one's sexuality and preponderance to sexual violence: age, education, spouse working status, wealth status, husband's alcohol consumption, women autonomy, decision-making, and freedom for mobility. This study suggests that empowering women with education, creating awareness regarding reproductive health, and addressing their socio-economic needs to help them achieve autonomy and derive decision-making power.

## Introduction

Gender-based violence against women is a significant public health concern and a embarrassing violation of human rights. Sexual violence occurs virtually across all regions and cultures [1] of the world with its varying definitions and degrees [2]. Globally, it is estimated that about one in three (33%) women have ever experienced either physical or sexual intimate partner violence or non-partner sexual violence in their life [3]. According to WHO, sexual violence can be defined as "any sexual act, attempt to obtain a sexual act, unwanted sexual comments or advances, or acts to traffic or otherwise directed against a person's sexuality using coercion, by any person regardless of their relationship to the victim, in any setting, including but not limited to home and work" [4]. Millions of women have been suffering from one or the other form of sexual violence worldwide irrespective of place of residence and geographical locations. Evidence shows that partner's lifetime prevalence of sexual violence ranged from six per cent in Japan to 59% in Ethiopia [5]. It ranges from 23.2% in high-income countries to 37.7% in the South-East Asian region [3]. A majority of the victims tend to avoid reporting these experiences due to one or the other reasons such as shame, reprisal, or deep-rooted gender inequity [6]. The fear of sexual violence among women is most likely restrict their freedom and occupational opportunities and affect their long-term psychological well-being.

The effect of violence on women's physical and mental health is detrimental, yet in many places worldwide, available services do not meet the requirements of victims. The consequences of sexual violence includes sexually transmitted infections (STIs) [7–9], signs of reproductive tract infection, unintended pregnancy, and non-use or inconsistent use of family planning methods [10–13]. However, its impact on the survivor's psychological health can be as grave as its physical bearing and maybe equally long-lasting [14, 15]. Deaths following sexual violence may result from suicide, HIV infection, or murder [5, 16].

The factors influencing a woman's risk of sexual violence are complex because it is deeply rooted in the inherent patriarchal system [17]. Previous pieces of evidence from different settings has attributed several risk factors to sexual violence among women including- teenage marriages, low knowledge of sexual matters, unaware of and unable to exercise sexual and reproductive rights, imbalanced gender norms, women's inability to negotiate sexual issues, and lack of alternative social support systems [1, 18–21]. It is found that women who marry at later ages are less likely to report coercive sexual experiences than women who marry in their teenage [18, 21–23]. It has also been reported that the partner's habit of alcohol consumption plays a critical role in violence against women [11, 24]. Further, infertility, husband's extramarital relations, intergenerational experience to violence, household economic pressure, and residing in the high crime-prone localities are positively associated with sexual violence [25, 26].

It has also been postulated that gender inequality in terms of financial control, decision-making regarding their healthcare may increase the risk of sexual violence [27–29]. Moreover, in highly patriarchal societies, such as India, where traditional gender paradigms exist, women have remained subordinate to men in almost all aspects of their lives in both custom and practice. Significant control exists on woman's sovereignty to follow their decisions and act on their well-being. Additionally, cultural, social and religious patterns in India collectively impose woman's lower status in family and society; and these may aggravate sexual violence [25, 30, 31]. It is also found that there exists a notion of assertion of sexual power to subjugate, given the unequal power dynamics between genders [32]. Further, sexual education in school and counselling related to sex and sexuality are still considered taboos in India [33].

Sexual activity among women presumably takes place within the wedlock and, out of it considered against the norms and sinful. Moreover, with the inherent nature of shyness and taboo associated with matters relating to sex, husbands generally see no problem in exercising some force when they desire sexual intercourse. Though evidence remains unclear, most of the literature available has come are from unrepresentative, small-scale studies [34, 35]. Therefore, this study aims to explore the various socio-economic and context-specific characteristics of husbands and wives on sexual autonomy and unwanted sexual experiences of currently married women in India.

It is well-known that sexual violence is a consequence of multiple factors which include an individual's (women's) and husband's characteristics, family, and social norms. Sexual violence against women can cause physical and mental distress to women, ranging from assaults, abuse, or deprivation in her life. Even though husbands/intimate partners are the most frequent perpetrators of domestic abuse, social norms and taboos in India play a significant role in making women victims. Fig 1 is a complex pathway that depicts the connections between the individual, socio-cultural, economic, and demographic factors that help to understand how sexual violence is played out and undermined in India. The factors involved in and their consequence to sexual violence are indicated with black arrows.

## Methods

### Data

For this study, we have utilized data from a cross-sectional study that was intended to bring significant evidence about health and family welfare and address critical issues from all Indian

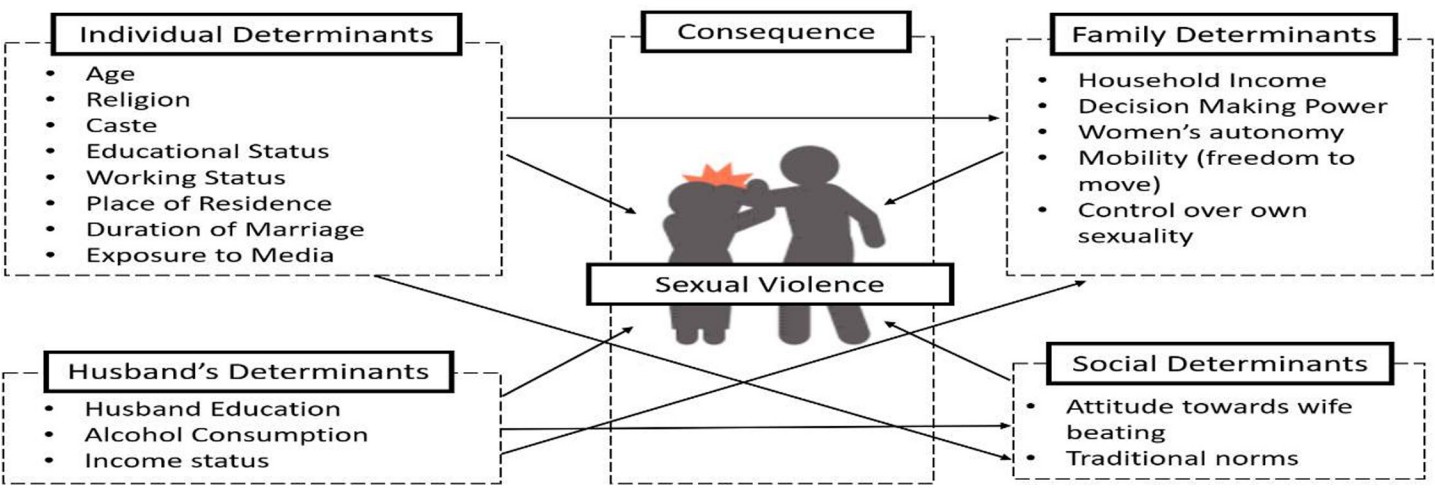

**Fig 1. Conceptual framework.**

States and Union territories. The National Family Health Survey-4 (NFHS-4) is a nationally representative survey and the fourth round was completed in 2015–16. All four survey questionnaires (Household Questionnaire, Woman's Questionnaire, Man's Questionnaire, and Biomarker Questionnaire) were surveyed using Computer Assisted Personal Interviewing (CAPI). The NFHS-4 survey implemented a stratified two-stage sample design to reach out to the survey households. We have utilized the couple file for our analysis in this study. A total of 63,696 couples were included in the analysis comprising of women of 15–49 years age and men of 15–54 years age [36].

## Statistical analysis

**Outcome variables.** The present study comprises two dependent variables: the first one is "control over sexuality," and the second is "sexual violence." The variable control over their sexuality was generated using three pieces of information: a) reason for not having sex: husband had a sexually transmitted disease (STI) b) reason for not having sex: husband had sex with other women and c) reason for not having sex: tired, not in the mood. All three variables were computed control over own sexuality (yes/no). The first category comprised of those who reported 'yes' for all three variables and those who answered 'no' in any one of the variables was considered as 'no'. The second dependent variable sexual violence was considered to have two categories–'yes' and 'no'.

**Predictor variables.** We have included predictor variables that are derived from across the literature and causal relationships with the control over one's own sexuality and sexual violence. The spousal educational status was generated through a continuous variable for schooling among husband and wife that was recoded as (both has no schooling, the wife was more educated, both equally educated and husband more educated) [37]. Spousal age gap was created through the continuous variable of age for husband and wife and was recoded as (husband and wife of the same age, wife older and husband older) [38]. Spousal working status was computed through the working status of husband and wife, and was recoded as (both not working, the only wife working, the only husband working and both working) [39]. Couple media exposure on family planning was created through family planning media exposure to husband and wife, and that was recoded as (no and yes). Religion was recoded as (Hindu, Muslim and others), caste (SC/ST and Non-SC/ST) [40], place of residence (urban and rural), wealth status (poorest, poorer, middle, richer and richest), regions (North, Central, East, Northeast, West and South). Husband alcohol consumption was recoded as (no and yes), husband education (no education, primary, secondary and higher), attitude towards wife-beating, autonomy and decision-making power (low, medium and high). Freedom to move (not alone and alone) [41].

## Statistical analysis

As the first step in the analysis, the study considered the control over one's own sexuality and sexual violence by selected background characteristics of the participants. This was followed by an estimation of the adjusted effects of the variables in the household characteristics, along with individual characteristics, on control over one's sexuality and sexual violence by using the binary logistic regression model. All the estimates and standard errors were adjusted for the multistage sampling design and clustering at the primary sampling unit and weighted at the State level to give unbiased results representative of the population. The statistical analysis was done in MS-Excel and STATA 13 software (Stata Corp, College Station, Texas).

## Results

The descriptive statistics for the target population are summarised in Table 1. It is assuring to learn that around 85 percent of the couples were educated with varied difference in the education level of the husbands and wives. Ninety-three percent of the wives were younger than their husbands. In the majority of the households (69%), only the husbands worked, while only two percent of the families had wives as the only earning member. Around 93% of the couples reported any kind of exposure to media. Nearly two-thirds of the couples were married for more than ten years. The remaining proportion was equally comprised of the couples married for 5–9 years and less than five years. Seventy-five percent of the couples were Hindus. Around 18 percent of the couples belonged to SC/ST category. The urban-to-rural proportion of these study participants was three to seven. However, the study participants were almost equally distributed in all five categories of wealth status. Geographically, western India (11%) had the least representation while the highest, around 25 percent of the couples, represented central India. It was found that 97 percent of the husbands among the target population did not consume alcohol. More than 80 percent of the men had attained some level of education; half of them obtained up-to secondary education. The difference in the distribution of the study participants was almost nil among the categories describing the attitude of the husband on physically beating their wives, women's autonomy and decision making, although women's freedom to move alone was slightly less than the counters.

Table 2: Bivariate analysis was used to understand the association between different socio-demographic covariates with the proportion of women not having control on their sexuality and experiencing sexual violence. The pattern of women's sexual violence varies with the spousal educational level. Women from the uneducated couple category experienced more sexual violence (10%) and had less control over their sexuality (34%) than the women of the other categories. As anticipated, among couples who had similar education levels, the women exercised more control over their own sexuality and had lower chances of experiencing sexual violence.

Age of the husband or wife did not bear any positive significance on the proportion of women who did not have control over their sexuality or experienced sexual violence. Irrespective of the working status of the woman, 30 percent of the women lacked control over their sexuality; working women with non-working husbands experienced the highest sexual violence (9%) among the rest. The women under the category 'couple with no media exposure' experienced higher sexual violence than their counterpart. However, media exposure among couples did not have any influence on the proportion of women having no control over their sexuality. The proportion of women not having control over their sexuality did not see many variations among the subcategories of the factors such as duration of marriage, religion, caste, women's autonomy and women's freedom to move alone (around 30%), however, as the years of marriage increased, it exposed the women to more sexual violence and less control over their sexuality. The proportion of women subjected to sexual violence was more or less the same in all religious groups. Women of other religious denominations reported better control over their sexuality. Sexual violence was more prevalent among the SC/ST women. Data also revealed that with the increase in wealth status of the family, sexual violence on women decreased and so did their self-control over sexuality increased. Surprisingly, women from the south India had the least control over their sexuality. On the other hand, women from the eastern zone experienced a higher rate of sexual violence. Women from rural areas were also more prone to sexual violence and less control over their sexuality.

The study also explored the role of a husband's habits and characteristics of women's control over sexuality and sexual violence. Husband's alcohol consumption had a direct influence on women's low control over sexuality and high exposure to sexual violence. An increase in

Table 1. Socio-demographic profile of the study participants (N = 63,696).

| Variables | Sample (N) | Percent (%) |
|---|---|---|
| **Spousal educational difference** | | |
| Both had no schooling | 9,245 | 14.51 |
| Wife more educated | 13,845 | 21.74 |
| Both equally educated | 9,056 | 14.22 |
| Husband more educated | 31,550 | 49.53 |
| **Spousal age gap** | | |
| Both same age or wife older | 4,374 | 6.87 |
| Husband Older | 59,322 | 93.13 |
| **Spousal working status** | | |
| Both not working | 4,386 | 6.89 |
| Only wife working | 1,238 | 1.94 |
| Only husband working | 43,710 | 68.62 |
| Both working | 14,362 | 22.55 |
| **Couple media exposure** | | |
| No | 4,525 | 7.1 |
| Yes | 59,171 | 92.9 |
| **Duration of marriage (years)** | | |
| 0–4 | 11,465 | 18 |
| 5–9 | 11,836 | 18.58 |
| 10+ | 40,395 | 63.42 |
| **Religion** | | |
| Hindu | 48,042 | 75.42 |
| Muslim | 8,302 | 13.03 |
| Others | 7,352 | 11.54 |
| **Caste** | | |
| SC/ST | 11,386 | 17.88 |
| Non-SC/ST | 52,310 | 82.12 |
| **Residence** | | |
| Urban | 18,944 | 29.74 |
| Rural | 44,752 | 70.26 |
| **Wealth Status** | | |
| Poorest | 11,395 | 17.89 |
| Poorer | 13,395 | 21.03 |
| Middle | 13,552 | 21.28 |
| Richer | 12,853 | 20.18 |
| Richest | 12,501 | 19.63 |
| **Regions** | | |
| North | 13,542 | 21.26 |
| Central | 15,630 | 24.54 |
| East | 10,420 | 16.36 |
| North East | 8,210 | 12.89 |
| West | 6,919 | 10.86 |
| South | 8,975 | 14.09 |
| **Husband alcohol consumption** | | |
| No | 61,900 | 97.18 |
| Yes | 1,796 | 2.82 |
| **Husband education** | | |

(*Continued*)

**Table 1.** (Continued)

| Variables | Sample (N) | Percent (%) |
|---|---|---|
| No | 11,747 | 18.44 |
| Primary | 10,025 | 15.74 |
| Secondary | 33,693 | 52.9 |
| Higher | 8,231 | 12.92 |
| **Attitude towards beating the wife** | | |
| Low | 21,232 | 33.33 |
| Medium | 21,331 | 33.49 |
| High | 21,133 | 33.18 |
| **Autonomy** | | |
| Low | 21,232 | 33.33 |
| Medium | 21,237 | 33.34 |
| High | 21,227 | 33.33 |
| **Decision making** | | |
| Low | 21,240 | 33.35 |
| Medium | 21,279 | 33.41 |
| High | 21,177 | 33.25 |
| **Freedom to move** | | |
| Not alone | 37,470 | 58.83 |
| Alone | 26,226 | 41.17 |

sexual violence and a decrease in control over one's sexuality was observed with a decrease in the education status of the husbands. From the analysis, it was found that the women experienced high sexual violence and no power to control their sexuality with husbands who resorted to physically beating their wives, less autonomy, low decision-making authority and limited freedom of movement.

## Regression results

Table 3 presents an adjusted odds ratio from the logistic regression to understand the association between women's sexual violence and women's control of their sexuality with various demographic and socio-economic contextual variables. In general, the results in Table 3 were substantiated by the odds ratio from the analysis.

The adjusted multivariate logistic regression model after controlling for the socio-economic factors shows that educated husbands were more likely to have control over their educated wives' sexuality (AOR: 0.88; CI:0.78–0.99) and this was statistically significant. Women having older husband were less likely to be having no-control over sexuality (AOR: 0.89; CI:0.83–0.95) and experienced sexual violence (AOR: 0.81; CI:0.7–0.95) and both were statistically significant. The study reveals that working women experienced higher odds of no-control over sexuality and sexual violence. Loss of control over one's sexuality is 20 percent more likely among the families where only women worked (AOR:1.2; CI:1.04–1.39) and her chances of experiencing sexual violence is 34 percent (AOR: 1.34; CI:1.02–1.76) more than the families with both husband and wife not working, both being statistically significant. The chances of experience of sexual violence was more likely among couples, when both are working (AOR: 1.37; CI:1.16–1.62) and was a statistically significant finding. A notable result as observed from the duration of the marriage factor is that, with increasing years of marriage, the likeliness of suffering from sexual violence increased. It is observed that Muslim women are more likely to have no control of sexuality (AOR: 1.39; CI:1.32–1.47), and other religious group women are

**Table 2. Percentage of women not having control over own sexuality and experienced sexual violence by background characteristics in India.**

| Variables | % of women not having control over own sexuality (N = 63,696) | Sexual violence (N = 47,514) |
|---|---|---|
| **Spousal educational difference** | | |
| Both had no schooling | 34.20 | 9.60 |
| Wife more educated | 31.51 | 5.89 |
| Both equally educated | 26.91 | 4.54 |
| Husband more educated | 28.27 | 6.18 |
| **Spousal age gap** | | |
| Both same age or wife older | 30.01 | 7.06 |
| Husband older | 29.64 | 6.30 |
| **Spousal working status** | | |
| Both not working | 28.51 | 6.88 |
| Only wife working | 30.71 | 8.70 |
| Only husband working | 29.62 | 5.65 |
| Both working | 29.97 | 8.03 |
| **Couple media exposure** | | |
| No | 30.00 | 8.88 |
| Yes | 29.63 | 6.17 |
| **Duration of marriage (years)** | | |
| 0–4 | 29.00 | 4.85 |
| 5–9 | 28.47 | 6.34 |
| 10+ | 30.17 | 6.73 |
| **Religion** | | |
| Hindu | 29.84 | 6.25 |
| Muslim | 30.59 | 6.82 |
| Others | 24.79 | 6.57 |
| **Caste** | | |
| SC/ST | 30.55 | 7.78 |
| Non-SC/ST | 29.43 | 5.98 |
| **Residence** | | |
| Urban | 28.79 | 4.88 |
| Rural | 30.12 | 7.14 |
| **Wealth Status** | | |
| Poorest | 31.04 | 10.48 |
| Poorer | 31.11 | 7.43 |
| Middle | 31.92 | 6.77 |
| Richer | 31.70 | 4.63 |
| Richest | 23.22 | 3.40 |
| **Regions** | | |
| North | 17.53 | 4.72 |
| Central | 18.58 | 7.38 |
| East | 31.51 | 9.44 |
| North East | 40.74 | 5.63 |
| West | 25.07 | 2.82 |
| South | 45.69 | 6.3 |
| **Husband alcohol consumption** | | |

(*Continued*)

**Table 2.** (Continued)

| Variables | % of women not having control over own sexuality (N = 63,696) | Sexual violence (N = 47,514) |
|---|---|---|
| No | 29.58 | 6.19 |
| Yes | 35.26 | 17.13 |
| **Husband education** | | |
| No | 34.40 | 9.06 |
| Primary | 31.99 | 8.03 |
| Secondary | 28.56 | 5.70 |
| Higher | 24.92 | 3.31 |
| **Attitude towards beating the wife** | | |
| Low | 35.02 | 9.30 |
| Medium | 28.51 | 5.40 |
| High | 24.67 | 3.88 |
| **Autonomy** | | |
| Low | 30.61 | 7.89 |
| Medium | 29.01 | 6.67 |
| High | 29.33 | 4.49 |
| **Decision making** | | |
| Low | 33.40 | 8.89 |
| Medium | 27.26 | 4.15 |
| High | 28.17 | 5.95 |
| **Freedom to move** | | |
| Not alone | 30.93 | 7.25 |
| Alone | 27.90 | 5.11 |

less likely to have no sexual control (AOR: 0.9; CI:0.84–0.96), both findings being statistically significant. The non-SC/ST women were 22 percent less likely to experience sexual violence than their counterparts. Similarly, urban women were also less likely to experience sexual violence (AOR: 0.83; CI:0.75–0.92). Looking at the economic status factor, we observe that women belonging to all other quintiles were less likely to have no control over their sexuality or experienced sexual violence, compared to the poorest index, and the odds ratio decreased with an increase in wealth quantile. Geographically, the women from southern part were more likely to not having control over own sexuality than the others (AOR: 3.79; CI:3.56–4.04) while, Eastern-zone women were more likely to experience sexual violence than the others, all of which were statistically significant. With the husband's alcohol consumption, the likelihood of women experienced sexual violence was higher (AOR: 1.58; CI:1.32–1.89) than those were non-alcoholic. The women having better-educated husbands were less likely to experience sexual violence (AOR: 0.90; CI:0.81–0.99). While husbands with only secondary education are less likely to have control of sexuality than with the husband with no education. Also, high or medium attitude towards wife-beating is less likely to be having no control over her sexuality or being suffering from sexual violence than the low attitude counterparts. The women's autonomy plays a major role in terms of sexual violence and control of sexuality, as with higher autonomy, the women were less likely to experience violence (AOR:0.81; CI:0.72–0.91). Women having medium or high decision-making power were less likely to have no control over their sexuality or being suffered by sexual violence, in comparison to the women having lower decision making power. The women who had the freedom to move alone were less likely to experience sexual violence and have no-control of sexuality (AOR: 0.77; CI:0.71–0.83 and

**Table 3. Adjusted odds ratio of women not having control over own sexuality and experienced sexual violence by background characteristics.**

| Variables | % of women not having control over own sexuality (N = 63,696) | Sexual violence (%) (N = 47,514) |
|---|---|---|
| **Spousal educational difference** | | |
| Both had no schooling | Ref. | Ref. |
| Wife more educated | 0.91(0.83,1.01) | 1.06(0.88,1.29) |
| Both equally educated | 0.88*(0.78,0.99) | 1.02(0.81,1.3) |
| Husband more educated | 0.96(0.87,1.07) | 1.02(0.82,1.27) |
| **Spousal age gap** | | |
| Both same age or wife older | Ref. | Ref. |
| Husband Older | 0.89*(0.83,0.95) | 0.81*(0.7,0.95) |
| **Spousal working status** | | |
| Both not working | Ref. | Ref. |
| Only wife working | 1.2*(1.04,1.39) | 1.34*(1.02,1.76) |
| Only husband working | 0.98(0.92,1.06) | 0.97(0.83,1.13) |
| Both working | 0.94(0.87,1.02) | 1.37*(1.16,1.62) |
| **Couple media exposure** | | |
| No | Ref. | Ref. |
| Yes | 0.96(0.89,1.03) | 1.05(0.92,1.21) |
| **Duration of marriage (years)** | | |
| 0–4 | Ref. | Ref. |
| 5–9 | 0.96(0.91,1.02) | 1.26*(1.1,1.44) |
| 10+ | 1.04(0.99,1.1) | 1.23*(1.09,1.39) |
| **Religion** | | |
| Hindu | Ref. | Ref. |
| Muslim | 1.39*(1.32,1.47) | 0.9(0.79,1.02) |
| Others | 0.90*(0.84,0.96) | 0.95(0.83,1.1) |
| **Caste** | | |
| SC/ST | Ref. | Ref. |
| Non-SC/ST | 1.00(0.95,1.05) | 0.78*(0.71,0.86) |
| **Residence** | | |
| Urban | Ref. | Ref. |
| Rural | 1.00(0.95,1.05) | 0.83*(0.75,0.92) |
| **Wealth Status** | | |
| Poorest | Ref. | Ref. |
| Poorer | 0.93*(0.88,0.99) | 0.84*(0.75,0.94) |
| Middle | 0.87*(0.81,0.92) | 0.85*(0.75,0.97) |
| Richer | 0.81*(0.76,0.87) | 0.71*(0.62,0.83) |
| Richest | 0.63*(0.58,0.68) | 0.63*(0.53,0.76) |
| **Regions** | | |
| North | Ref. | Ref. |
| Central | 0.79*(0.74,0.84) | 1.29*(1.13,1.46) |
| East | 1.29*(1.21,1.38) | 1.85*(1.62,2.11) |
| North East | 2.66*(2.48,2.86) | 1.36*(1.16,1.59) |
| West | 1.59*(1.48,1.71) | 0.63*(0.52,0.76) |
| South | 3.79*(3.56,4.04) | 1.05(0.9,1.21) |

(*Continued*)

**Table 3.** (Continued)

| Variables | % of women not having control over own sexuality (N = 63,696) | Sexual violence (%) (N = 47,514) |
|---|---|---|
| **Husband alcohol consumption** | | |
| No | Ref. | Ref. |
| Yes | 1.03(0.92,1.14) | 1.58*(1.32,1.89) |
| **Husband education** | | |
| No | Ref. | Ref. |
| Primary | 0.95(0.86,1.05) | 1.00(0.81,1.22) |
| Secondary | 0.90*(0.81,0.99) | 0.91(0.74,1.12) |
| Higher | 0.90(0.80,1.02) | 0.62*(0.47,0.81) |
| **Attitude towards beating the wife** | | |
| Low | Ref. | Ref. |
| Medium | 0.78*(0.75,0.82) | 0.60*(0.55,0.65) |
| High | 0.75*(0.72,0.79) | 0.49*(0.44,0.55) |
| **Autonomy** | | |
| Low | Ref. | Ref. |
| Medium | 0.93*(0.88,0.97) | 0.99(0.9,1.09) |
| High | 1.03(0.97,1.08) | 0.81*(0.72,0.91) |
| **Decision making** | | |
| Low | Ref. | Ref. |
| Medium | 0.79*(0.75,0.82) | 0.50*(0.45,0.55) |
| High | 0.78*(0.74,0.82) | 0.69*(0.63,0.76) |
| **Freedom to move** | | |
| Not alone | Ref. | Ref. |
| Alone | 0.88*(0.85,0.91) | 0.77*(0.71,0.83) |

*if p<0.05; Ref.: Reference.

SC/ST: Scheduled Caste/Scheduled Tribe; %:percentage.

AOR: 0.88; CI:0.85–0.91) than the women who are exempted, respectively, all of which were statistically significant.

## Discussion

The prevalence and severity of gender-based violence vary within the country and communities, depending on the culture, tradition and social norms. Falling under the bracket of intimate partner violence, sexual violence against women is the most sensitive and least reported violence due to taboos, stigma, or inherent gender norms. Sexual violence is nothing but a violation of fundamental human rights, specifically violation of basic sexual and reproductive health rights across the globe. In spite of numerous laws being enacted to address the issue, not much of an impact is evident in women's lives or practice autonomy. Therefore, the present paper aims to assess the status of sexual independence and prevalence of sexual violence against women and enumerate the factors predicting the sexual autonomy and sexual violence against women among currently married women of India using the data from the fourth round of National Family Health Survey [36].

The study found that very close to one-third of married women in India were denied their sexual and reproductive health rights. However, the women are either not allowed to exercise

control over their sexuality and/or had ever experienced any form of sexual violence by their intimate partner in their lifetime. The study explores the various socio-economic and regional characteristics, spousal characteristics, and specific characteristics related to husbands and wives individually on the key outcomes of concern of this article.

## Spousal characteristics

Universally, education is considered a factor of empowerment for women [42, 43]. This is evident among equally educated spouses and women with any level of education that promoted better control over sexuality among women and reduced sexual violence. It is expected as education is highly related to knowledge and practice of sexual behaviour. Also, educated women are more to take their own decisions and object to men's dominating behaviours, and better equip themselves to negotiate sex [44]. Further, men with some schooling also reportedly thought rationally in any act. The study groups of women with older husbands, families with women as the only breadwinners and both the spouses working, increasing duration of the marriage, and couple with no media exposure on family planning, contributed either to women's reduced control over her sexuality or increased experience of sexual violence of varying order or both. The study further indicates that working women were more likely to experience sexual violence and have no control over their sexuality than the non-working comparable women's group. It may be because they might be more likely to challenge their intimate partner decision or because their husbands perceive a risk to their authority [45–49]. Besides social and cultural context, male disapproval of working wives may explain why not-working women are likely to experience intimate partner violence. The duration of the marriage is significantly associated with sexual violence against women. This translates that the women in difficult relationships of long durations are less bothered about social stigma and are more open to acknowledging intimate partner violence.

## Socio-economic and regional characteristics

The prevalence of sexual violence and loss of control over one's sexuality among women was more among women from the Muslim religion, from the SC/ST category, south and east region of India, rural areas, and women from poorer economic classes. The expression of reproductive health rights was better among women with improved financial status.

Generally, the caste group is considered a proxy of low socio-economic status in India [50–52]. It has been seen that the knowledge, awareness and literacy rate are inadequate among the individuals belonging to SC/ST caste. The household's wealth status was significantly negatively associated with no control over sexuality and sexual violence against women. Economic status is an indicator of social disadvantage. Hence, it may contribute to the risk of sexual abuse for females. Moreover, probably women from wealthier households are more sensitive to public view. They are more likely to provide socially expected reactions to survey investigators and hide their suffering than women belonging to more impoverished families [53].

Interestingly, the study asserted that women belonging to the western and southern regions were more likely to report no control over their sexuality. However, this can be because of the given evidence of geographic variations in female autonomy [54] and social inequality [55], where the regional patterns of social norms play a critical role.

## Husband characteristics

The study analyzed the impact of a husband's habits and characteristics on a woman's control over sexuality and sexual violence. Husband's alcohol consumption, lower educational status of husbands and husband's attitude that 'beating their wives is acceptable' had a direct

influence on women's no control over their sexuality and exposure to sexual violence against them. The consumption of alcohol has continuously emerged as a risk factor for violence against women [27] and consistent across various settings [56]. Alcohol consumption drives as a situational element that aggregates the probability of abuse by decreasing embarrassments, prohibition, and unclear decisions and ruining an individual's ability to interpret cues [57].

Moreover, from the present study, it is evident that the 'wife-beating is not right' attitude significantly reduced the risk of women not having control over their sexuality and sexual violence. Women who disagree with the justification of wife-beating have a lower chance of occupying inferiority relative to men, which probably makes them less susceptible to sexual violence by an intimate partner. The factors correlated with wife-beating reflect women's helplessness and deprivation-their lack of education, lack of control over resources, and lack of alternatives to early marriage [58–61].

## Wife characteristics

The study reveals that women experienced sexual violence, no power to exercise control over their sexuality with less autonomy, low decision-making authority and faced restrictions to move alone out of the house.

The research also confirms the significant negative association between decision-making power and women's no control over their sexuality and sexual violence among Indian women. From recent published literature, we find that women who had a bank account had reported lower chances of encountering sexual abuse as compared to their counterparts (OR 0.874, p < .001) [62, 63]. The association between freedom to go out of the house alone and lower risk of experiencing sexual violence and exercising better control over own sexuality can be translated by the fact that a woman's control over resources may augment her capacity to exercise choice. The benefit of having control over resources may give her the ability to balance the costs and benefits of substitute uses of economic, social resources so that resources can be managed in the most effective mode. Women having the freedom of movement may have more gender-egalitarian beliefs, and therefore, may be less vulnerable to sexual violence and have greater control over their sexuality.

The findings endorse that along with other socio-economic and demographic factors, women's autonomy, attitude towards wife-beating, women's decision-making power, and freedom of movement may need to be deliberated as the significant socio-cultural determinants for plummeting the chance of women's no control on her sexuality and sexual violence among Indian women.

## Conclusion

The study highlights the factors that are associated with women's control over their own sexuality and sexual violence and these include—age, education, spouse working status, wealth status, husband's alcohol consumption, women's autonomy, decision-making, and freedom to move. The study provides a canvas depicting the determinants/restraints of the socio-economic, various individual, and spousal characteristics on the Indian women's power to exercise control over her sexuality and experience of sexual violence. It also identifies the need to empower women with education, creating awareness of reproductive health, address the socio-economic needs to help them achieve autonomy and decision-making power. Along with this, there is also a need to ensure necessary actions are to be implemented and effective to eliminate social evils such as alcohol addiction and disband the attitude that it is okay to beat wives, in order to achieve healthy spousal relationship and build a healthy society.

## Author Contributions

**Conceptualization:** Gyan Chandra Kashyap, Shobhit Srivastava.

**Data curation:** Shobhit Srivastava.

**Formal analysis:** Shobhit Srivastava.

**Investigation:** Shobhit Srivastava.

**Methodology:** Shobhit Srivastava.

**Software:** Shobhit Srivastava.

**Supervision:** Gyan Chandra Kashyap.

**Writing – original draft:** Gyan Chandra Kashyap, Madhumita Bango, Subhojit Shaw.

**Writing – review & editing:** Bal Govind, Veena R.

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
