## [Decision Letter · Decision Letter 0]

5 May 2021

PONE-D-21-06643

A true face of Indian married couples: effect of age and education on control over own sexuality and sexual violence

PLOS ONE

Dear Dr. Shobhit,

Thank you for submitting your manuscript to PLOS ONE. After careful consideration, we feel that it has merit but does not fully meet PLOS ONE’s publication criteria as it currently stands. Therefore, we invite you to submit a revised version of the manuscript that addresses the points raised during the review process.

We look forward to receiving your revised manuscript.

Kind regards,

Russell Kabir, PhD

Academic Editor

PLOS ONE

Journal Requirements:

Reviewers' comments:

Reviewer's Responses to Questions

**Comments to the Author**

1. Is the manuscript technically sound, and do the data support the conclusions?

Reviewer #1: Yes

Reviewer #2: Yes

2. Has the statistical analysis been performed appropriately and rigorously? 

Reviewer #1: Yes

Reviewer #2: Yes

3. Have the authors made all data underlying the findings in their manuscript fully available?

Reviewer #1: Yes

Reviewer #2: Yes

4. Is the manuscript presented in an intelligible fashion and written in standard English?

Reviewer #1: Yes

Reviewer #2: No

5. Review Comments to the Author

Reviewer #1: The manuscript is technically sound and using NFHS data it depicts the very important domain of sexuality and sexual violence among Indian couples. The topic fulfils the dearth of couple related picture of sexual violence in India and how age and education of couple is associated with sexual violence. However, I have certain comments:

1. Specify total sample in the method section of the abstract.

2. Provide AOR in result section of the abstract.

3. Introduction is well written however a conceptual framework is needed.

4. Please do elaborate the data part and provide citations also.

5. Outcome variable section need a little elaboration

6. Provide some useful explanation and citation for explanatory variables in variable description section.

7. In the result section some changes are needed for example in regression based results provide CI in the result description also e.g., [AOR: 1.90; CI: 1.44-2.90], write the word “significantly more likely or significantly less likely”.

8. Please do check the paper for grammatical mistakes and provide the references in Vancouver style i.e., in big brackets [1], [2-4], [3, 7, 9] etc.

Reviewer #2: The article highlights a very sensitive and important topic “sexual intimate partner violence IPV and control over sexual activity”, the study focuses on the association between several factors “socio-economic, demographic factors, spousal characteristics and special characteristics” and risk of sexual IPV and control over sexual activity. The argument made in the study is valid and the study findings support similar findings in the field. The association is plausible, and the study design and methods are feasible as it uses secondary data already published. The material is all reasonable, reliable, and prepared. The methodology and analysis are explained in detail. Appropriate statistical tests were used with adjusted odds ratio, and the results section reflected on the statistically significant findings. Therefore, repetition of the study is possible, making the findings reliable. Discussion explains the results, however more references need to be added to some statements (explained below).

Comments:

Although the title and the aim mentioned in the introduction part of the abstract implies that the study focus on age and education as variables affection sexuality control and sexual violence, however the study covers more socio-economic and demographic factors; suggestion to edit that part to reflect more on the study. Also, the introduction in the abstract and the aim in the last part of the introduction hints that the study would cover policies and programs for prevention in India, those were not cover in the study, and need to be removed.

Suggest adding more keywords to cover socio-economic, demographic factors, and spousal characteristics and special characteristics.

The abstract need rephrasing especially the introduction, methodology, conclusion parts (need to reflect more on the conclusion at the end of the article).

line 120: Can you reference such studies “unrepresentative small-scale studies”

“Though evidence remains unclear, whichever has come are from unrepresentative, small-scale studies.”

The aim of the study Line 121: the aim seems different of that mentioned in the abstract, no mention of studying variable associations, and the study does not address policies and programs. Suggestion use aim in line: 281 here instead

“Therefore, the present study examines the status of sexual autonomy and unwanted sexual experiences among currently married women in India and also aims to address policies and programs for its prevention in India.”

Line 128: can you add the abbreviation here “NFHS-4” and reference.

Line 132: combine with line 133

Line 154: Caste “SC/ST” Maybe add a reference refer to schedule castes or schedule tribes.

Line 179: “three to seven” delete “is”

Line 194: “the results were better among the equally educated group” please rephrase for better understanding.

Line 204: (around 30)? percent?

Line 285- line 310: please reference.

Line: 332 and line 341 can you add discussion with other relevant studies with similar results if present?

6. PLOS authors have the option to publish the peer review history of their article (what does this mean?). If published, this will include your full peer review and any attached files.

Reviewer #1: No

Reviewer #2: **Yes: **Angi ALRADIE-MOHAMED

---

## [Author Response · Author response to Decision Letter 0]

7 Jun 2021

Reviewer #1: The manuscript is technically sound and using NFHS data it depicts the very important domain of sexuality and sexual violence among Indian couples. The topic fulfils the dearth of couple related picture of sexual violence in India and how age and education of couple is associated with sexual violence. However, I have certain comments:

Comment 1: Specify total sample in the method section of the abstract.

Response: We have included the information in the method section of the abstract. 

Comment 2: Provide AOR in result section of the abstract.

Response: We have inserted the AORs. 

Comment 3: Introduction is well written however a conceptual framework is needed.

Response: Thanks for praising the research. We have included the conceptual framework with the description in the introduction section. 

Comment 4: Please do elaborate the data part and provide citations also.

Response: We have made the suggested changes. 

Comment 5: Outcome variable section need a little elaboration.

Response: We have explained in a better way. 

Comment 6: Provide some useful explanation and citation for explanatory variables in variable description section.

Response: We have made the necessary changes and included the citations in the description of the explanatory variables. 

Comment 7: In the result section some changes are needed for example in regression-based results provide CI in the result description also e.g., [AOR: 1.90; CI: 1.44-2.90], write the word “significantly more likely or significantly less likely”.

Response: We have incorporated the suggested changes in the revised manuscript. 

Comment 8: Please do check the paper for grammatical mistakes and provide the references in Vancouver style i.e., in big brackets [1], [2-4], [3, 7, 9] etc.

Response: We have incorporated the suggested changes in the revised manuscript. 

 

Reviewer #2: The article highlights a very sensitive and important topic “sexual intimate partner violence IPV and control over sexual activity”, the study focuses on the association between several factors “socio-economic, demographic factors, spousal characteristics and special characteristics” and risk of sexual IPV and control over sexual activity. The argument made in the study is valid and the study findings support similar findings in the field. The association is plausible, and the study design and methods are feasible as it uses secondary data already published. The material is all reasonable, reliable, and prepared. The methodology and analysis are explained in detail. Appropriate statistical tests were used with adjusted odds ratio, and the results section reflected on the statistically significant findings. Therefore, repetition of the study is possible, making the findings reliable. Discussion explains the results, however more references need to be added to some statements (explained below).

Response: Thank you so much for praising the research article. 

Comment 1: Although the title and the aim mentioned in the introduction part of the abstract implies that the study focus on age and education as variables affection sexuality control and sexual violence, however the study covers more socio-economic and demographic factors; suggestion to edit that part to reflect more on the study. Also, the introduction in the abstract and the aim in the last part of the introduction hints that the study would cover policies and programs for prevention in India, those were not cover in the study, and need to be removed.

Response: We have incorporated the suggested changes and removed the said statement. 

Comment 2: Suggest adding more keywords to cover socio-economic, demographic factors, and spousal characteristics and special characteristics.

Response: We have included three more key word in the existing list. 

Comment 3: The abstract need rephrasing especially the introduction, methodology, conclusion parts (need to reflect more on the conclusion at the end of the article).

Response: We have made the necessary changes in the abstract and conclusion section of the manuscript. 

Comment 4: line 120: Can you reference such studies “unrepresentative small-scale studies”

“Though evidence remains unclear, whichever has come are from unrepresentative, small-scale studies.”

Response: We have included two references. 

• Babu BV, Kar SK. Domestic violence against women in eastern India: a population-based study on prevalence and related issues. BMC public health. 2009 Dec;9(1):1-5.

• Mahapatro M, Gupta RN, Gupta V. The risk factor of domestic violence in India. Indian journal of community medicine: official publication of Indian Association of Preventive & Social Medicine. 2012 Jul;37(3):153.

Comment 5: The aim of the study Line 121: the aim seems different of that mentioned in the abstract, no mention of studying variable associations, and the study does not address policies and programs. Suggestion use aim in line: 281 here instead. “Therefore, the present study examines the status of sexual autonomy and unwanted sexual experiences among currently married women in India and also aims to address policies and programs for its prevention in India.”

Response: We have incorporated the suggested modifications.

Comment 6: Line 128: can you add the abbreviation here “NFHS-4” and reference.

Response: Included the abbreviation. 

Comment 7: Line 132: combine with line 133.

Response: Incorporated the changes. 

Comment 8: Line 154: Caste “SC/ST” Maybe add a reference refer to schedule castes or schedule tribes.

Response: Inserted the reference in the line 154.

Waghmare, K. S. (2020). Scheduled Caste and Scheduled Tribe (Prevention of Atrocities) Act, 1989.

Comment 9: Line 179: “three to seven” delete “is”.

Response: We have incorporated the suggested deletion. 

Comments 10: Line 194: “the results were better among the equally educated group” please rephrase for better understanding.

Response: We have rephrased the sentence. 

As anticipated, couples who had similar education levels, in this case, women having more control over their own sexuality and lower chances of experiencing sexual violence.

Comments 11: Line 204: (around 30)? percent?

Response: Yes, integrated 

Comments 12: Line 285- line 310: please reference.

Response: We have included following references. 

• Sundaram MS, Sekar M, Subburaj A. Women empowerment: role of education. International Journal in Management & Social Science. 2014;2(12):76-85.

• Stromquist NP. Education as a means for empowering women. Rethinking empowerment: Gender and development in a global/local world. 2002 Sep 20:22-38.

• Tripathi N. Does family life education influence attitudes towards sexual and reproductive health matters among unmarried young women in India?. Plos one. 2021 Jan 25;16(1):e0245883.

• Sanneving L, Trygg N, Saxena D, Mavalankar D, Thomsen S. Inequity in India: the case of maternal and reproductive health. Global health action. 2013 Dec 1;6(1):19145.

• Nayar KR. Social exclusion, caste & health: a review based on the social determinants framework. Indian Journal of Medical Research. 2007 Oct 1;126(4):355.

• Chanana K. Accessing higher education: the dilemma of schooling women, minorities, Scheduled Castes and Scheduled Tribes in contemporary India. Higher Education. 1993 Jul;26(1):69-92.

Comments 13: Line: 332 and line 341 can you add discussion with other relevant studies with similar results if present?

Response: We have included the following references and discussed with the findings.

• Madan M. Understanding attitudes toward spousal abuse: Beliefs about wife-beating justification amongst men and women in India. Michigan State University. Criminal Justice; 2013.

• Jejeebhoy SJ, Cook RJ. State accountability for wife-beating: the Indian challenge. The Lancet. 1997 Mar 1;349:S10-2.

• Fulu E, Warner X, Miedema S, Jewkes R, Roselli T, Lang J. Why Do Some Men Use Violence Against Women and How Can We Prevent It? Quantitative Findings from the United Nations Multi-country Study on Men and Violence in Asia and the Pacific. 2013, Bangkok: UNDP, UNFPA, UN Women and UNV

• Rahman M, Nakamura K, Seino K, Kizuki M. Does gender inequity increase the risk of intimate partner violence among women? Evidence from a national Bangladeshi sample. PLoS One. 2013 Dec 23;8(12):e82423.

• Shabnam S. Sexual Violence and Women Empowerment in India: Findings from a Nationally Representative Sample Survey. 2021

• Zegenhagen S, Ranganathan M, Buller AM. Household decision-making and its association with intimate partner violence: Examining differences in men's and women's perceptions in Uganda. SSM-population health. 2019 Aug 1;8:100442.

---

## [Editor Report · Decision Letter 1]

18 Jun 2021

A true face of Indian married couples: effect of age and education on control over own sexuality and sexual violence

PONE-D-21-06643R1

Dear Dr. Srivastva,

We’re pleased to inform you that your manuscript has been judged scientifically suitable for publication and will be formally accepted for publication once it meets all outstanding technical requirements.

Kind regards,

Russell Kabir, PhD

Academic Editor

PLOS ONE
---

## [Editor Report · Acceptance letter]

25 Jun 2021

PONE-D-21-06643R1 

A true face of Indian married couples: effect of age and education on control over own sexuality and sexual violence 

Dear Dr. Srivastava:

I'm pleased to inform you that your manuscript has been deemed suitable for publication in PLOS ONE. Congratulations! Your manuscript is now with our production department. 

Kind regards, 

on behalf of

Dr. Russell Kabir 

Academic Editor

PLOS ONE